# Remaining Useful Life Prediction of Lithium-Ion Batteries Using Neural Networks with Adaptive Bayesian Learning

**DOI:** 10.3390/s22103803

**Published:** 2022-05-17

**Authors:** Karkulali Pugalenthi, Hyunseok Park, Shaista Hussain, Nagarajan Raghavan

**Affiliations:** 1Engineering Product Development Pillar, Singapore University of Technology & Design, 8 Somapah Road, Singapore 487372, Singapore; nagarajan@sutd.edu.sg; 2Department of Information Systems, Hanyang University, Seoul 133791, Korea; hp@hanyang.ac.kr; 3Computational Intelligence Group, A*STAR Institute of High-Performance Computing (IHPC), Singapore 138632, Singapore; hussains@ihpc.a-star.edu.sg

**Keywords:** lithium-ion batteries, particle filters, neural networks, remaining useful life, prognostics

## Abstract

With smart electronic devices delving deeper into our everyday lives, predictive maintenance solutions are gaining more traction in the electronic manufacturing industry. It is imperative for the manufacturers to identify potential failures and predict the system/device’s remaining useful life (RUL). Although data-driven models are commonly used for prognostic applications, they are limited by the necessity of large training datasets and also the optimization algorithms used in such methods run into local minima problems. In order to overcome these drawbacks, we train a Neural Network with Bayesian inference. In this work, we use Neural Networks (NN) as the prediction model and an adaptive Bayesian learning approach to estimate the RUL of electronic devices. The proposed prognostic approach functions in two stages—weight regularization using adaptive Bayesian learning and prognosis using NN. A Bayesian framework (particle filter algorithm) is adopted in the first stage to estimate the network parameters (weights and bias) using the NN prediction model as the state transition function. However, using a higher number of hidden neurons in the NN prediction model leads to particle weight decay in the Bayesian framework. To overcome the weight decay issues, we propose particle roughening as a weight regularization method in the Bayesian framework wherein a small Gaussian jitter is added to the decaying particles. Additionally, weight regularization was also performed by adopting conventional resampling strategies to evaluate the efficiency and robustness of the proposed approach and to reduce optimization problems commonly encountered in NN models. In the second stage, the estimated distributions of network parameters were fed into the NN prediction model to predict the RUL of the device. The lithium-ion battery capacity degradation data (CALCE/NASA) were used to test the proposed method, and RMSE values and execution time were used as metrics to evaluate the performance.

## 1. Introduction

Maintenance of electronic devices, physical equipment and systems is imperative for ensuring successful functioning, minimal downtime, reduced unprecedented maintenance costs, and prolonging the life of the device/system. Maintenance strategies are broadly categorized as preventive and predictive maintenance strategies. Although preventive maintenance adopts a conventional approach of regular scheduled maintenance protocols, predictive maintenance strategies are preemptive methods. With the availability of affordable sensor systems and advances in machine learning algorithms, predictive maintenance approaches promise economic benefits both for the manufacturers and end-users.

The goal of predictive maintenance strategies is to predict the remaining useful life (RUL) of the device/system. RUL prediction of devices can be realized through two methods, namely, model-based methods and data-driven methods. Model-based methods employ physical or statistical models which best capture the degradation of the desired system. However, obtaining such models requires extensive experimental work and an in-depth understanding of the underlying failure mechanisms. These models are effective until the system is upgraded or changed. Commonly used model-based prognostic approaches are Kalman filters (KF) [1,2], Extended Kalman filters (EKF) [3,4], and Particle filters (PF) [5,6]. These methods either use empirical models or incremental state-space representations of the underlying governing partial differential equations (PDE). On the other hand, data-driven methods use machine learning based approaches to characterize the failure of the desired device/system. Machine learning methods use pattern recognition to learn features from raw sensor data and identify failure patterns. Machine learning based data-driven methods successfully used for developing prognostic algorithms include regression methods [7,8], support vector machine (SVM) [9,10], relevance vector machine (RVM) [11,12], Bayesian networks (BN) [13,14], hidden Markov model (HMM) [15], and artificial neural networks (ANN) [16,17]. Among the different machine learning methods, neural network (NN) based methods are preferred by researchers due to their versatility and adaptive parameter optimization capabilities. To elaborate, NN based methods can address uncertainties in the system, such as measurement uncertainties introduced by noisy sensor data and prediction model uncertainties caused by variations in operating conditions. However, shallow neural network architectures, such as feedforward neural networks and radial bias function neural networks, face challenges in predicting the RUL with good confidence due to its dependency on large training datasets. Additionally, the complexity of the network architecture increases for highly nonlinear systems, which severely affects parameter optimization and further leads to overfitting issues.

Several deep learning models are proposed in the literature to address optimization and overfitting issues, such as convoluted neural networks (CNN) [18,19], recurrent neural networks (RNN) [20,21], deep belief networks (DBN) [22], and long short-term memory networks (LSTM) [23,24]. Although deep learning models are efficient methods for RUL prediction due to their ability to learn features on-the-fly, they are computationally expensive. For instance, combining convoluted features across all time steps in CNN is time-consuming and inhibits its application in processing large-scale time-series data. Similarly, RNN based methods are sequential wherein long-term information has to sequentially fed through all cells before being processed, subsequently causing vanishing gradient issues. LSTM based methods, on the other hand, require a large amount of memory bandwidth for processing large-scale time-series data.

Additionally, there have been several attempts made to develop hybrid prognostic algorithms combining different model-based and data-driven methods to overcome the limitations arising due to dependency on accurate physical models and large amount of failure data. To name a few, Ma et al. [25] developed a hybrid prognostic model combining CNN and LSTM. The authors used CALCE and NASA datasets to evaluate their performance and used nearly 50% of data from each battery for the purpose of training. The training dataset size was optimized by using false nearest neighbor method. Even though the authors obtained good prediction accuracy, the RUL prediction can be performed only at mid/late degradation cycles which limits their applicability to safety-critical devices wherein the failures even in the early degradation cycles can lead to catastrophic results. Similarly, Wu et al. [26] developed a hybrid prognostic approach combining NN and particle filters wherein they had used the NN model equation as the system degradation model in the PF algorithm. In this case, the authors had assumed that they have the run-to-failure data for all the batteries available and used curve fitting results of each battery as the initial parameter guess for the PF algorithm. This limits the generalization capability of their proposed approach. This calls for the need to optimize shallow neural network models by regularizing network parameters and complexity to provide an efficient and computationally inexpensive framework for prognostic studies appropriate for real-time applications.

## 2. Training Algorithms for Neural Networks

Artificial neural network models have been successfully applied in prognostic studies [16,17] due to their ability to model non-linearities in degradation data and generalization capabilities. However, choosing an appropriate NN model is an arduous task as its optimization capabilities rely heavily on network architecture, training algorithms, and initial values of connection weights and bias. The NN model learns patterns from training data during the training phase and uses that information to produce the desired output. The NN model parameters (weights and bias) are modified to minimize the error between the predicted value and the desired output. A suitable backpropagation algorithm does the process of adjusting the network parameters for minimizing the error value. Thus, choosing an efficient training algorithm is imperative as it directly affects the performance of the network model. The commonly used NN training algorithms are gradient descent and Levenberg–Marquardt algorithms. However, the major drawback of using backpropagation-based training algorithms is that if the error values are multimodal, the algorithm becomes trapped at local minima. Additionally, improper/random initial parameter configuration aids the algorithm to settle at incorrect local minima.

To overcome the disadvantages of backpropagation algorithms, several evolutionary algorithms have been proposed in the literature. Gudise et al. [27] proposed the particle swarm optimization (PSO) algorithm for training a simple feedforward NN model. The authors compared the performance of PSO with gradient descent (GD) algorithm. The results clearly showed that the convergence rate of PSO was much better than GD. Similarly, Karaboga et al. [28] proposed an artificial bee colony (ABC) algorithm to train a feedforward NN model. The performance of the algorithm was compared with genetic algorithm (GA) and GD. The results proved that ABC algorithm does not become trapped at local minima. However, the computational cost of the proposed method was considerably high as it took a very high number of epochs for convergence. Additionally, in all of the algorithms mentioned above, the initial parameter values were deterministic, so when applied to time-series prediction models, they would only yield a single-point estimate for the future values.

This work proposes using a Bayesian inference-based training algorithm to identify the NN model parameters and provide probabilistic RUL estimates over a single point estimate. Here, we resort to using particle filters for training a feedforward MLP model wherein the NN model equation is used in the PF algorithm as the incremental state transition function for predicting the system’s future state. The PF algorithm performs state estimation to deduce the model parameters best representing the training dataset. Further, the PF estimated model parameters are used to configure the initial connection weight and bias values of the MLP network for other devices being monitored to predict the system’s future state and subsequently for RUL estimation.

Usage of PF for NN training is promising due to its ability to easily capture the highly nonlinear and non-Gaussian statistics by using different weights for different samples. It is to be noted that the particle filter weights are different from the NN weight values. However, the weighted sampling approach of PF leads to a particle collapse wherein the particle weights accumulate over a small fraction of particles known as particle degeneracy. If left unaddressed, particle degeneracy eventually leads to particle impoverishment wherein the particle weights are accumulated on a single particle, and the rest of the particles have zero weights. Particle impoverishment affects the performance of the NN model in the prognosis stage as the network model becomes configured with incorrect initial parameter weight and bias values. Although particle degeneracy can be controlled by adopting suitable resampling strategies, it is challenging to overcome particle impoverishment. In order to reduce particle impoverishment, we proposed *particle roughening* as a weight regularization method.

The rest of the paper is organized as follows. Section 3 describes the two different lithium-ion battery capacity degradation datasets used in this work. Section 4 describes the methodology used in this work—description of multilayer perceptron, particle filters, and the proposed RUL estimation framework. Section 5 summarizes the results and discussion, and conclusions and future work are discussed in Section 6.

## 3. Degradation Datasets

Two different lithium-ion battery capacity degradation datasets from CALCE and NASA repository are used in this work for testing the performance of the proposed method [29,30].

### 3.1. CALCE Dataset

Accelerated aging tests were performed for a set of prismatic cells (CS) with LiCoO_2_ cathode of 1.1 Ah capacity rating. The CS cells were charged and discharged repeatedly till the cells reached their End-of-Life (EoL). The cells were subjected to a charging profile using constant current/constant voltage protocol. The current rate was maintained at 1C till the voltage reached 4.2 V.

 The charge was sustained at 4.2 V till the charging current dropped to 0.05A. The failure threshold for the CS cells was set to be at 0.88 Ah. The capacity degradation curves for three CS cells from CALCE repository are shown in Figure 1a.

### 3.2. NASA Dataset

The second dataset used in this work is from NASA Ames Prognostic Center of Excellence [26]. In this dataset, the LiCoO_2_ cathode cells with a rated capacity of 2.1 Ah were used for generating the battery capacity degradation data. Unlike the CALCE dataset, the NASA batteries were tested under random discharge currents rather than constant discharge currents. The charge/discharge cycles termed Random Walk (RW) cycling was performed wherein the current profile for both charging and discharging were changed every 5 min with a current value randomly selected from 0–4 A. The randomized loading conditions were applied to generate degradation data to simulate more realistic operating conditions. The failure threshold was set to be at 1 Ah, and the battery capacity degradation curves are shown in Figure 1b.

## 4. Methodology

### 4.1. Multilayer Perceptron

The most popular NN architecture used is the feedforward multilayer perceptron (MLP). We resort to using two-layer MLP in this work. The NN architecture represents the degradation state of the device as a function of time. The input node and output node represent the input and output variables, respectively. The input node is fed with time in cycles (battery) as the input to obtain the corresponding degradation state—capacity (battery). The hidden layer connects the input and output node and is represented by nonlinear nodes called hidden neurons. Each layer transmits information forward through an activation function. A sigmoid activation function is used between the input and hidden layer, followed by a linear activation function between the hidden and output layer. The sigmoid activation function in terms of NN parameters (weights and bias) is expressed as:(1)$$h_{i}=\frac{1}{1+e^{-\left(w_{i}^{\left(1\right)}\times k+b_{i}^{\left(1\right)}\right)}}$$
where *w_i_*^(1)^ and *b_i_*^(1)^ is the input weight and bias corresponding to the *i*th hidden neuron, *k* is the time index, *i* = 1, *…*, *M* is the number of hidden neurons, and *h_i_(.)* is the tan-sigmoid activation function corresponding to the input layer. The predicted capacity/light output at the output node can be represented as
(2)$$g\left(\left(w,b\right),k\right)=f(\overset{M}{\underset{i=1}{\sum }}\left(h\left(\left(k\times w_{i}^{\left(1\right)}+b_{i}^{\left(1\right)}\right)w_{i}^{\left(2\right)}\right)+b_{i}^{\left(2\right)}\right)$$
where *g((w,b),k)* is the output of the MLP network, *w_i_*^(2)^ and *b_i_*^(2)^ represent the weight and bias values at the hidden layer, and *h(.)* is the tan-sigmoid activation function between the input and hidden layer as shown in Equation (1).

#### Choice of Network Architecture

One of the significant challenges with NN is the choice of network architecture. Choosing an appropriate number of hidden layers and neurons dramatically affects the performance of the NN model, especially while handling prognostic applications. During the training phase, the NN model parameters (*w_i_* and *b_i_*) are optimized to minimize the prediction error on the training patterns. Minimal error values denote a stable network, whereas high error values reflect overfitting. Using more hidden neurons might cause the network to overfit and lead to significant deviations in the predicted values. As there are no standard approaches available in the literature to deduce the optimum number of hidden neurons, we chose to use the Bayesian Information Criterion (*BIC*) to fix the suitable number of hidden neurons. *BIC* is a widely used metric for statistical model selection owing to its computational efficiency and simplicity [31,32]. *BIC* of a model can be evaluated as
(3)BIC=−2×LL+ln(N)×q
where *LL* refers to the log-likelihood function of the model, *N* is the size of the training dataset, and *q* is the number of parameters to be estimated by the model. Additionally, the *BIC* penalizes a model based on the number of estimated parameters; hence, a complex model with higher neurons would be penalized and yield a poor (higher) *BIC* value. The model with the minimum *BIC* values is chosen as the best model for the purpose. The *BIC* analysis is shown in Figure 2. From Figure 2, it can be inferred that the NN model with three hidden neurons is suitable for the battery degradation dataset.

### 4.2. Adaptive Bayesian Learning

#### 4.2.1. Particle Filters

In this work, the PF algorithm is used as a state estimation method to deduce the optimum NN model parameter values (weights and bias). PF is a recursive Bayesian algorithm in which the system is represented by a state-space model comprising of a state transition model and measurement model as shown below: (4)$$x_{k}=f(x_{k-1},\theta )+\omega_{k-1}$$
(5)$$z_{k}=q(x_{k},\theta )+\varepsilon_{k}$$
where *x_k_* and *x_k−1_* represent the current and previous degradation state of the system, *z_k_* represents the available test data at *k*th time instant, *k* is the time index(cycles/hours), θ represents the vector with MLP model parameters (*w_i_* and *b_i_*),  ω*_k−1_* is the process noise, and $\varepsilon_{k}$ is the measurement noise present in the system. *f(.)* represents an incremental model of the state transition function, and *q(.)* represents the measurement function. In this work, the measurement function used is *g((w,b),k)* represented by Equation (2). The implementation of the PF algorithm is explained below:a. Particle Initialization—At *k* = 1 time instant, the initial prior distribution is populated based on prior knowledge of the system model parameters as shown in Figure 3a. In this case, the curve fitting coefficients corresponding to one device from each device datasets are used. The initial prior probability density function *p(x*_0_*)* is then sampled into weighted particles.b. Particle Update—Whenever new measurement data are available for prediction, the weights of the particles are recursively updated based on the likelihood function, as shown below
(6)$$p(z_{k}|x_{k}^{i},\theta_{k}^{i})=\frac{1}{\sqrt{2\pi }\sigma }exp\left[-\frac{1}{2}\frac{(z_{k}-q{\left(x_{k}^{i}\right)}^{2}}{\sigma^{2}}\right]$$
(7)$$w_{k}^{i}=\frac{p(z_{k}|x_{k}^{i},\theta_{k}^{i})}{\sum_{i=1}^{j}p(z_{k}|x_{k}^{i},\theta_{k}^{i})}$$
where $p(z_{k}|x_{k}^{i},\theta_{k}^{i})$ is the likelihood function with $\theta_{k}^{i}$ representing the MLP network parameter at *k*th time instant, *z_k_* is the available test data, *x_k_* is the current system degradation state, and *w_i_^k^* represents the particle weights. It is to be noted that the particle weights of the PF algorithm are different from the MLP network parameter weights.c. Particle Resampling—During particle update, the weights accumulate over a few particles, and the rest of the particles carry negligible weights after a few iterations as shown in Figure 3b. In order to enhance diversity amongst the particles, the smaller weight particles are replaced with large weight particles by a process called particle resampling. Thus, the basic idea of resampling is to maintain all the samples/particles at the same weight.d. State Estimation—Finally, the degradation state of the system at (*k* + 1)th time instant is evaluated based on the new set of weighted particles. The process is repeated for all available measurement data, and the posterior distribution at the current step becomes the prior distribution for the next step.

#### 4.2.2. Weight Regularization Methods

Although a very good candidate for non-linear degradation prognosis in general with non-Gaussian noise components, the significant drawbacks of the PF algorithm are the particle degeneracy phenomenon followed by particle impoverishment. During particle updates, the particles with negligible weights are replaced by large weight particles. After few iterations, small weights will cease to exist and only large weight particles are present in the distribution. This phenomenon is termed particle degeneracy, and particle impoverishment is a severe case of particle degeneracy wherein all but one particle is eliminated during resampling, i.e., a single particle carries all the weights, as shown in Figure 3c. This dramatically affects the diversity of particles, thus constraining the evolution of model parameters and subsequently affects the accuracy of predictions.

Choosing an appropriate resampling strategy is one of the simplest methods to address particle degeneracy. In this work, we have applied three different resampling strategies based on our previous work in Ref. [33] to improve the adaptive Bayesian learning framework used in this study. The three resampling strategies considered are Multinomial Resampling (MR), Stratified Resampling (StR), and Systematic Resampling (SyR). The schematic representation for all three resampling strategies is depicted in Figure 4a. The weights of five particles after normalization are shown for illustration in Figure 4a. The length of the rectangles depicts the weights of the particles.

 MR is a random search approach where *N* independent particles are randomly selected from the particle distribution. StR divides the population into equal segments called strata, and particles are randomly selected from each stratum. SyR, on the other hand, is an extension of stratified resampling wherein one particle from each stratum from a fixed location is selected during resampling. Thus, SyR is a more deterministic approach compared to the other two resampling strategies.

Similarly, to address particle impoverishment, we use particle roughening as a weight regularization technique [34]. Although resampling strategies try to diversify the particle weight distribution, particle roughening, on the other hand, is a compensation technique. If particle impoverishment has occurred despite choosing an appropriate resampling strategy, one way to reduce it would be to redistribute the weights centered around one particle by jittering their values. For jittering, a small random noise (roughening noise) is added to the resampled particles. The roughening noise is small Gaussian jitter with zero mean and constant covariance. The covariance matrix is obtained from the standard deviation of the system degradation state as shown in Equation (8):(8)$$\sigma_{K}^{i}=\left(\sigma_{r1},\sigma_{r2}\right)$$
(9)$$\sigma_{r1}=\sqrt{-2\sigma_{1}^{2}\times \ln\left(L_{K}^{i}\right)}$$
(10)$$\sigma_{r2}=\sqrt{-2\sigma_{2}^{2}\times \ln\left(L_{K}^{i}\right)}$$
where $\;\sigma_{K}^{i}$ is the standard deviation of the Gaussian jitter with $\sigma_{r1}$ being the standard deviation corresponding to the lower limit ($\sigma_{1}^{2})$ and $\sigma_{r2}\;$ corresponding to the upper limit ($\sigma_{2}^{2})$ and $L_{K}^{i}$ represents the likelihood function corresponding to Equation (6). The standard deviation limits used in this work are summarized in Table 1. When particle diversity improves, the distribution of model parameters is decentralized, and hence the prediction performance of the NN model improves.

### 4.3. Remaining Useful Life Estimation

The proposed approach, as shown in Figure 5, is split into two stages: Stage A—Adaptive Bayesian learning framework with weight regularization and Stage B—Prognosis using NN. Stage A is further split into three steps—Data Preprocessing, Particle Filters, and Weight Regularization, the descriptions of which are explained below.

Data Preprocessing—An appropriate NN model is chosen based on the *BIC* analysis described in previous sections.Particle Filters—The chosen model is used as the measurement function in the particle filter algorithm to recursively update the model parameters using the available degradation data from the training dataset.Weight Regularization—To overcome the particle degeneracy and impoverishment issues, suitable resampling strategies, and roughening methods are adopted for weight regularization. Additionally, resampling/roughening at every time step is unnecessary as it introduces additional variance in the posterior distribution. Hence, Effective Sample Size (ESS) is introduced to regulate the resampling/roughening process. If the ESS of the distribution is lower than a predefined threshold *N_T_*, then the particles are resampled/roughened. This reduces unnecessary additional computational load. The parameters obtained at the final time step for the training dataset is the PF estimated MLP network parameters.

In Stage B, a new MLP network architecture is configured with PF estimated network parameters, which are the initial parameter values for corresponding weight and bias at the input and hidden layers. The available degradation data of other devices in the lot (test dataset—device 2 and 3) is fed to the MLP model as input. The MLP is trained using the LM algorithm for the test dataset. The informed PF estimated initial parameter values help to prevent local minima encountered in the backpropagation learning algorithm of neural networks. The trained NN model is used to predict the future degradation state of the device till end-of-life.

During backpropagation, there is a possibility that the network parameters can go astray and distort the prediction traces. A suitable success criterion is essential to filter out outliers from the degradation traces. Two success criteria are incorporated into the framework. One is to eliminate traces that lie beyond the 2σ limits of the majority of the prediction traces. The second criterion is to eliminate traces which has a monotonically increasing trend utterly different from the actual degradation data used in this work. The predicted EoL is evaluated based on the successful traces, and eventually, the RUL of the device is calculated. The performance of the proposed framework is evaluated using the percentage of successful iterations, Root Mean Squared Error (RMSE), Relative Accuracy (RA), and computational time as metrics.

## 5. Results and Discussion

### 5.1. RUL Estimation Using Different Resampling Strategies

The prediction results for CALCE battery degradation dataset are discussed in this section. The battery CS-36 was used for training the model, and batteries CS-37 and CS-38 were chosen as the test datasets. As per the *BIC* analysis discussed in Section 3, the network architecture which best represents the CALCE dataset consists of an MLP network with three hidden neurons. The corresponding model equation was used for generating the curve fitting coefficients which was, in turn, used to populate the initial prior distribution in the PF algorithm. The three neuron NN model equation (as per Equation (2)) was used as the measurement function in the PF algorithm, which recursively updates the model parameters for the training dataset (CS-36). The PF algorithm encounters weight decay issues during particle updates, as illustrated in Figure 3b. Hence, suitable resampling strategies were adopted to sort the weight decay issues. 50 sets of model parameter values from the posterior distribution of the PF algorithm were fed into the MLP model as initial parameter configuration value. Available data points from the test dataset were fed as input to the MLP model for prognosis.

The prediction results for CS-37 using multinomial resampling (MR) are shown in Figure 6b. Based on the assumed success criteria, the green traces in the Figure 6b are considered unsuccessful and eliminated for RUL prediction. The magenta traces correspond to prediction traces with RMSE values below 1% and thus is considered the best traces amongst the 50 repetitions. It can be observed from Figure 6b that only about 78% of the prediction traces were successful. The corresponding posterior model parameter distributions obtained after state estimation is shown in Figure 6a. A significant variance in the posterior distributions indicates that the weight decay issues of the PF algorithm remain unresolved for MR. The prediction results for CS-37 using stratified resampling (StR) are shown in Figure 7b. In this case, we opted to stratify the weight distribution into N_s_/2 times, where N_s_ is the total number of particles used in the PF algorithm. During resampling, at least one particle from each stratum was picked, thereby improving particle diversity. Although StR yields a success rate of about 90%, the posterior distribution of model parameters shown in Figure 7a still shows negatively skewed distributions with no diversity. This indicates that if the PF algorithm fails to converge close to the true parameter values, then the prediction performance can be affected adversely.

Further, systematic resampling (SyR) was adopted to improve the accuracy of predictions. The number of successful iterations jumped to 96%, and the posterior parameter distributions were found to be more spread out, as shown in Figure 8a,b, respectively. The results clearly depict that resolving the weight decay issues in the PF algorithm regularizes the NN weights and bias values and, in turn, helps to improve the prediction accuracy. 

The performance metrics comprising of RMSE, computational time and percentage of successful iterations are summarized in Table 2. It can be inferred from Table 2 that SyR performs the best amongst the different resampling strategies explored in this work and also eliminates the particle degeneracy issues. Since SyR is deterministic in nature, it proves to be better than the other two resampling strategies. However, systematic resampling does not overcome particle impoverishment. To elucidate, we have shown the particle weight distribution plots in Figure 9a–d, while using SyR. The total number of time steps till the actual EoL for CALCE dataset is 112. The particle weight distributions at 76th, 80th, 96th, and 112th time step (last time step) is shown in Figure 9. The particle weights start to lose their diversity around 80th time step and subsequently suffers severe particle impoverishment around the last iteration. In order to over this issue, particle roughening strategies were adopted.

### 5.2. RUL Estimation Using Particle Roughening Method

For particle roughening, the standard deviation of the Gaussian jitters is the key influencing factor for improving particle diversity. Based on Ref. [34], three different sigma values were used in literature to simulate the jittering effect as shown in Table 1, and Sigma-2 values were found to be the best-performing ones. The sigma limits were chosen based on the admissible values of measurement noise to be present in the system under consideration. Hence, we adopted Sigma-2 for generating the Gaussian jitter to be added to the resampled particles. The particles were resampled using SyR and the resampled particles were added with a small Gaussian jitter with zero mean and standard deviation corresponding to Sigma-2. 

However, roughening comes with an additional computational cost as shown in Table 2. Particle roughening takes about 8 to 10 times more time than SyR. The big advantage though is that the proposed method eliminates particle impoverishment. The posterior parameter distributions and the prediction traces for CS-37 using particle roughening strategy are shown in Figure 10a,b, respectively. The number of successful iterations improves to 98% with just one unsuccessful iteration. Additionally, the prediction traces are more intact and closer to the true values shown by the black line in Figure 9a. Moreover, the posterior distributions of parameters are more spread out clearly indicating a better particle diversity.

The proposed weight regularization method has advantages over other evolutionary algorithms, such as genetic algorithm and particle swarm optimization algorithm, which are highly computationally expensive approaches. To the best of our knowledge, the proposed adaptive Bayesian learning with weight regularization is the first of its kind to be used to optimize the MLP parameters for prognostic applications.

To check the robustness of the proposed method, we also applied the proposed prognostic approach to the NASA battery degradation dataset. RW10 was used for training the model, and battery RW11 was the test dataset. The performance results are again summarized in Table 2. The prognostic metrics used in this work are root mean squared error (RMSE), relative accuracy (RA) and computational time. The RMSE and RA values deduced in this work were evaluated using the following standard expressions in Equations (11) and (12). In Equation (11), n is the size of the training dataset, T is the prediction starting point index value and k is the number of cycles.
(11)$$RMSE=\sqrt{\frac{1}{n}\overset{n}{\underset{k=T}{\sum }}{\left(x_{predicted}-x_{true}\right)}_{k}^{2}}$$
(12)$$Relative\;Accuracy\;\left(RA\right)=1-\frac{\left|Predicted\;RUL-True\;RUL\right|}{True\;RUL}$$

The results indicate that the proposed prognostic framework with weight regularization outperforms the standard resampling strategies in the literature. The parameter distributions from the adaptive Bayesian learning framework and the corresponding degradation traces for NASA battery dataset are shown in Figure 11, Figure 12, Figure 13 and Figure 14. Additionally, the comparison between predicted RUL and true RUL for both the datasets using the different resampling and roughening methods at three different prediction starting points are shown in Figure 15 and Figure 16, respectively, in terms of the RA metric.


From Figure 15 and Figure 16, it can be inferred that the accuracy of the resampling strategy adopted is reflected in the closeness of the predicted RUL to the true RUL value of the device under consideration The true RUL for both the datasets at different prediction starting points are represented by the magenta, red and cyan dotted lines. For both the datasets, particle roughening method performs the best with minimum RUL error value. Additionally, the variance in the RUL distribution can be inferred from the height of the box-plot shown in Figure 15 and Figure 16. Thus, the results clearly show that the predictions results obtained using particle roughening method are both accurate as well as precise (relatively low variance in predicted RUL compared to most other common resampling methods).

### 5.3. Comparison of Prediction Results with Previous Works in the Literature

In order to evaluate the performance of our proposed method, we have compared our prediction results with two other relevant methods available in the literature. The prediction results obtained from Refs. [26,35] are used for comparison. The authors in Ref. [26] had developed a hybrid prognostic algorithm wherein the NN degradation model was used in the particle filter algorithm as the degradation model and further extrapolated in future for prognosis. In this work, the run-to-failure data of each battery were needed to determine the initial parameter guess which were fed to the PF algorithm. Similarly, Ref. [35] is one of our earliest works wherein we had developed a particle filter trained neural network model for prognosis. However, the efficacy of the proposed method was limited due to particle degeneracy and impoverishment issues. Hence, we adopted suitable weight regularization techniques in this work to overcome those disadvantages. The prognostic metric used in all the three work is the prediction RMSE value which are summarized in Table 3 below. It is evident from Table 3 that our proposed method here with weight regularization techniques incorporated into the hybrid framework performs better than the previous works from the literature.

## 6. Conclusions

In this work, we proposed an adaptive Bayesian learning framework to train MLP models for prognostic application on multiple electronic devices. The proposed adaptive Bayesian learning framework uses a particle filter algorithm for state estimation, wherein the weight decay issues commonly encountered in PF algorithms adversely affect the convergence of MLP weight and bias values and lead to poor prognostic performance. Hence, we adopted three different resampling strategies and particle roughening approaches into the PF framework. These strategies enable weight regularization in the MLP model used for prognosis. The proposed method was tested out on CALCE and NASA battery degradation datasets with high non-linearity and non-monotonicity. The prediction results clearly showed that systematic resampling helped improve particle diversity in the PF algorithm and subsequently helped eliminate particle degeneracy. Additionally, including the suitable particle roughening strategies in systematic resampling helps to eliminate particle impoverishment. The results imply that the proposed adaptive Bayesian learning framework with weight regularization helps the model parameters converge closer to the true values, prevent local minima problems, and helps to improve the generalization capabilities of NN models.

In the future, we intend to modify the proposed framework and incorporate physics informed loss functions into the MLP architecture. The purpose of including a physics-based loss function would be to apply the framework for devices with highly noisy data (and sparse good data scenarios as well) and underlying failure mechanisms with hidden physics. Although physics informed machine learning approaches are the state-of-the-art, the computational framework proposed thus far still have discrepancies in their convergence rates and suffer from vanishing backpropagation gradients. Thus, introducing a Bayesian inference-based training process can help to overcome the optimization and convergence issues.

## Figures and Tables

**Figure 1 sensors-22-03803-f001:**
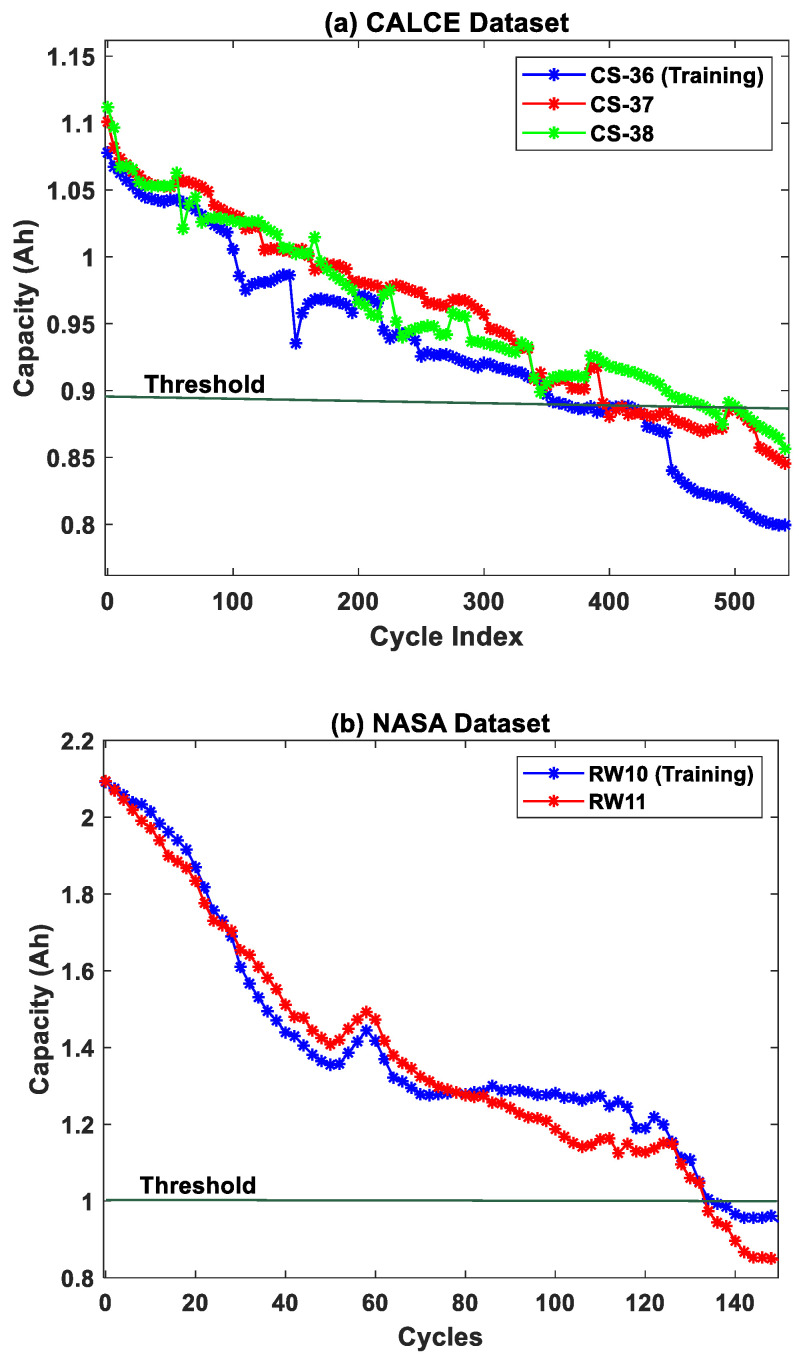
The battery capacity degradation datasets from the (**a**) CALCE and (**b**) NASA repositories.

**Figure 2 sensors-22-03803-f002:**
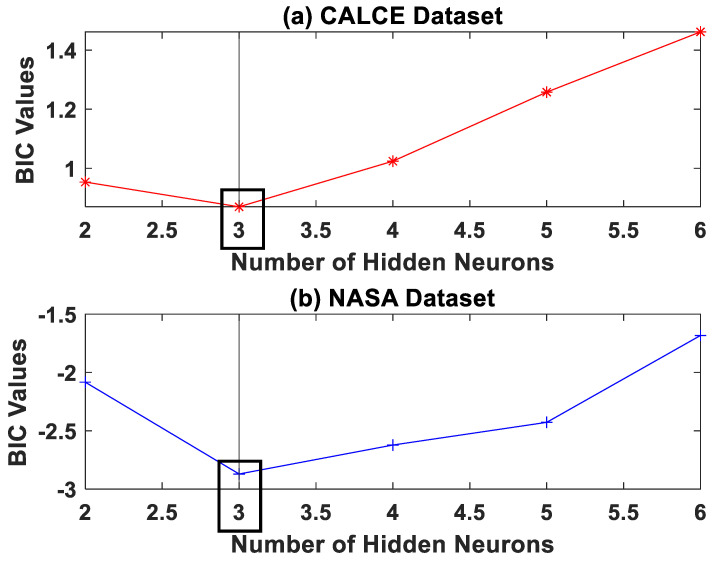
The *BIC* analysis for model selection for the two lithium-ion battery capacity degradation datasets—(**a**) CALCE and (**b**) NASA datasets.

**Figure 3 sensors-22-03803-f003:**
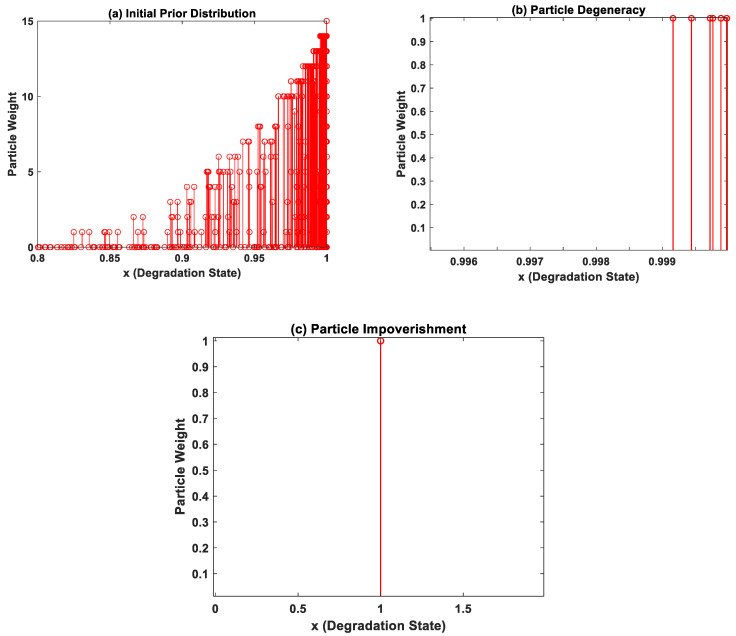
The particle weight distributions obtained while executing the PF algorithm: (**b**) weight distribution depicting particle degeneracy; and (**c**) weight distribution depicting particle impoverishment.

**Figure 4 sensors-22-03803-f004:**
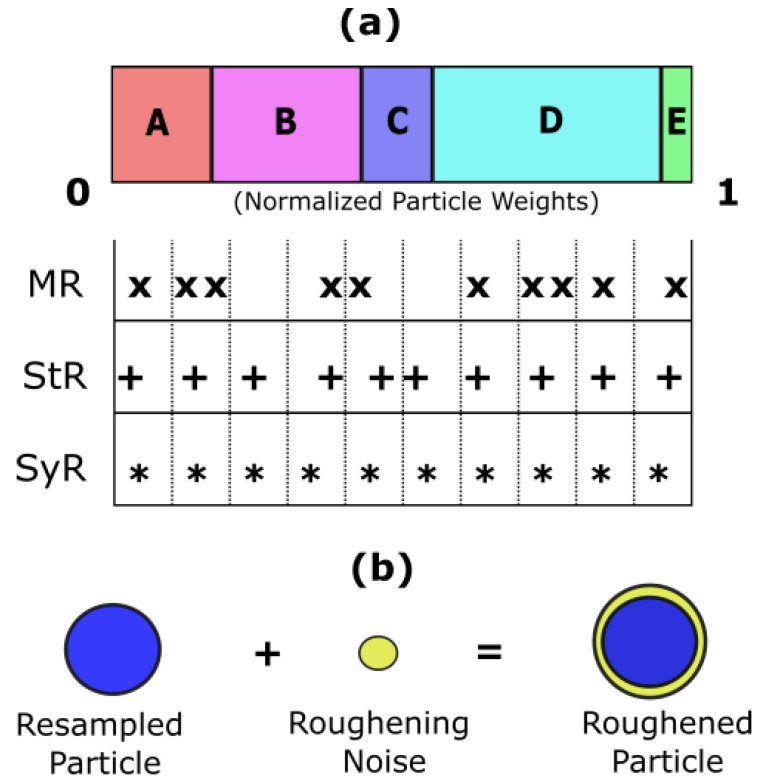
(**a**) The schematic representation of different resampling strategies used in this work—Multinomial Resampling (MR), Stratified Resampling (StR), and Systematic Resampling (SyR); and (**b**) the schematic representation of particle roughening weight regularization method.

**Figure 5 sensors-22-03803-f005:**
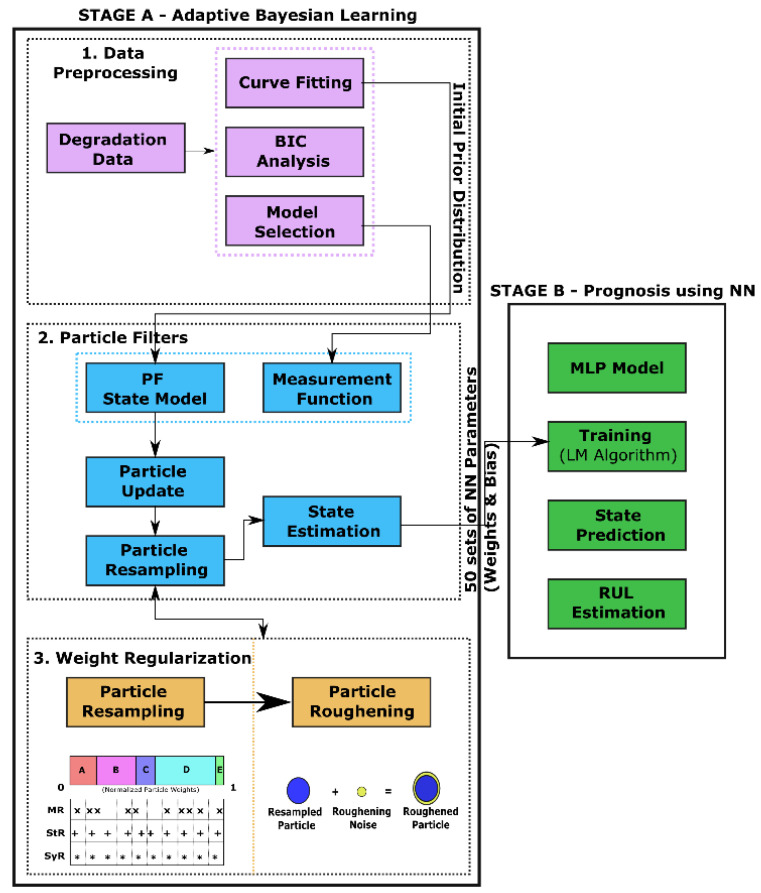
The schematic of the proposed prognostic framework using an adaptive Bayesian learning framework with weight regularization for training the MLP network model.

**Figure 6 sensors-22-03803-f006:**
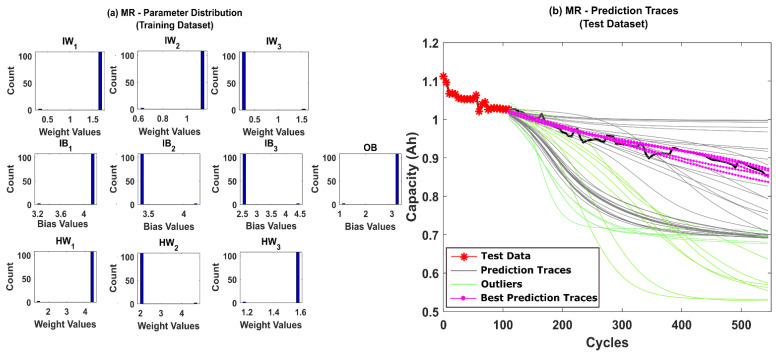
(**a**) The posterior distribution of MLP network parameters estimated by the adaptive Bayesian learning framework using MR and (**b**) the degradation prediction traces for CALCE battery (CS-37) using MR. The prediction traces for 50 repetitions are shown using the gray lines, the green lines represent outliers, and the magenta lines represent the traces with minimum RMSE values.

**Figure 7 sensors-22-03803-f007:**
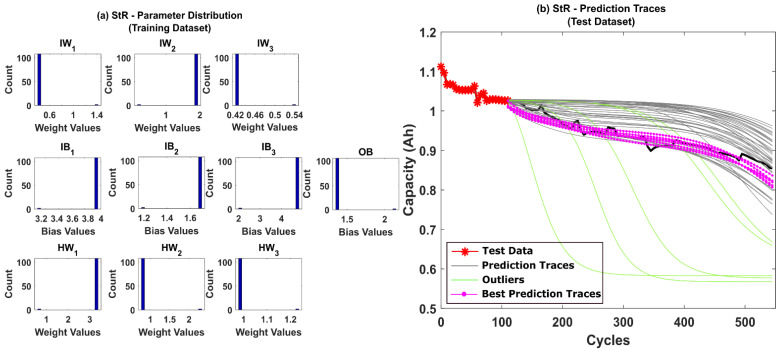
(**a**) The posterior distribution of MLP network parameters estimated by the adaptive Bayesian learning framework using StR and (**b**) the degradation prediction traces for CALCE battery (CS-37) using StR. The prediction traces for 50 repetitions are shown using the gray lines, the green lines represent outliers, and the magenta lines represent the traces with minimum RMSE values.

**Figure 8 sensors-22-03803-f008:**
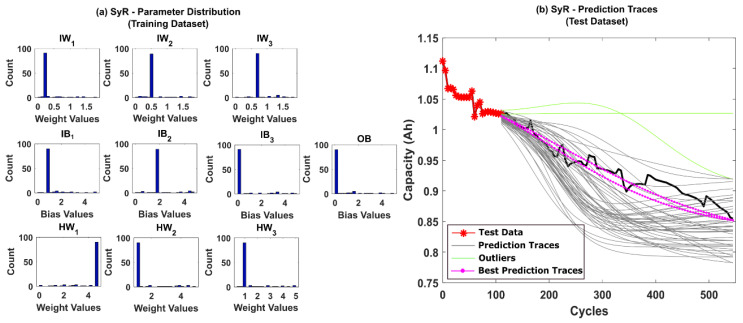
(**a**) The posterior distribution of MLP network parameters estimated by the adaptive Bayesian learning framework using SyR and (**b**) the degradation prediction traces for CALCE battery (CS-37) using SyR. The prediction traces for 50 repetitions are shown using the gray lines, the green lines represent outliers, and the magenta lines represent the traces with minimum RMSE values.

**Figure 9 sensors-22-03803-f009:**
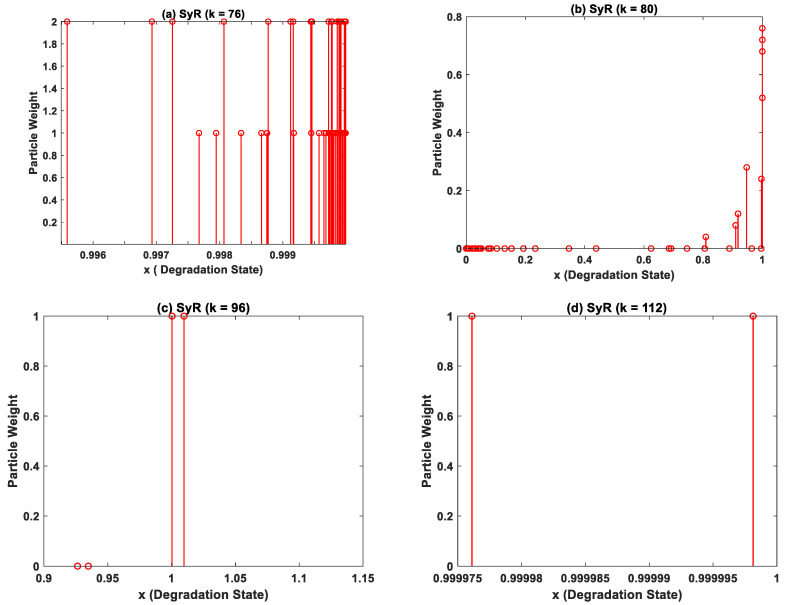
The particle weight distribution obtained during execution of particle filter algorithm using systematic resampling (SyR) at the (**a**) 76th (**b**) 80th (**c**) 96th, and (**d**) 112th time step. The particle weight distribution in (**d**) depict particle impoverishment despite using adopting robust resampling strategies.

**Figure 10 sensors-22-03803-f010:**
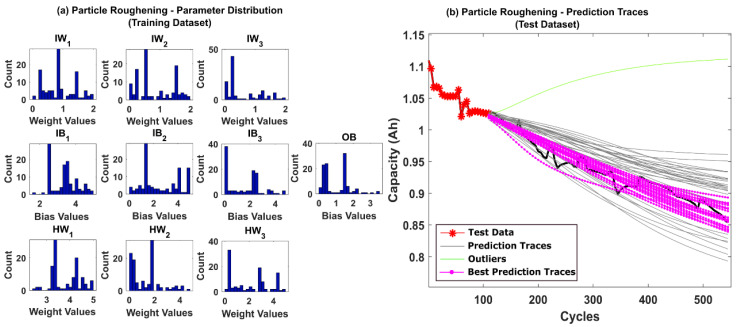
(**a**) The posterior distribution of MLP network parameters estimated by the adaptive Bayesian learning framework using particle roughening and (**b**) the degradation prediction traces for CALCE battery (CS-37) using particle roughening. The prediction traces for 50 repetitions are shown using the gray lines, the green lines represent outliers, and the magenta lines represent the traces with minimum RMSE values.

**Figure 11 sensors-22-03803-f011:**
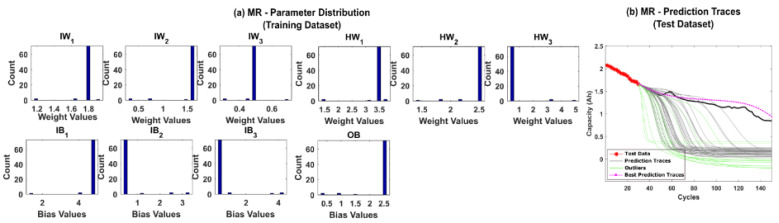
(**a**) The posterior distribution of MLP network parameters estimated by the adaptive Bayesian learning framework using MR and (**b**) the degradation prediction traces for NASA battery (RW-11) using MR. The prediction traces for 50 repetitions are shown using the gray lines, the green lines represent outliers, and the magenta lines represent the traces with minimum RMSE values.

**Figure 12 sensors-22-03803-f012:**
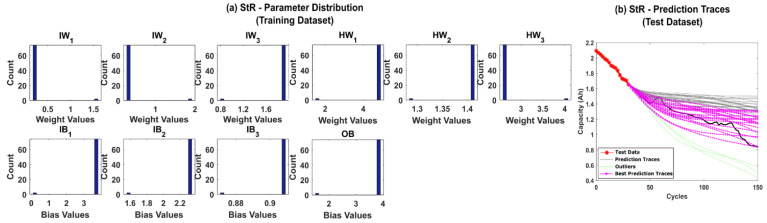
(**a**) The posterior distribution of MLP network parameters estimated by the adaptive Bayesian learning framework using StR and (**b**) the degradation prediction traces for NASA battery (RW-11) using StR. The prediction traces for 50 repetitions are shown using the gray lines, the green lines represent outliers, and the magenta lines represent the traces with minimum RMSE values.

**Figure 13 sensors-22-03803-f013:**
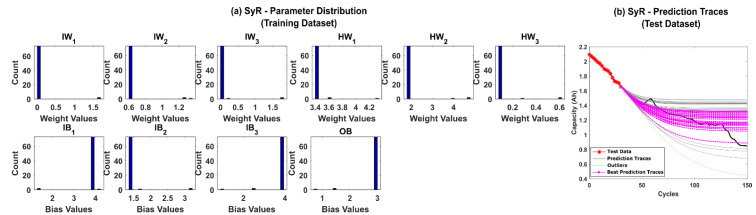
(**a**) The posterior distribution of MLP network parameters estimated by the adaptive Bayesian learning framework using SyR and (**b**) the degradation prediction traces for NASA battery (RW-11) using SyR. The prediction traces for 50 repetitions are shown using the gray lines, the green lines represent outliers, and the magenta lines represent the traces with minimum RMSE values.

**Figure 14 sensors-22-03803-f014:**
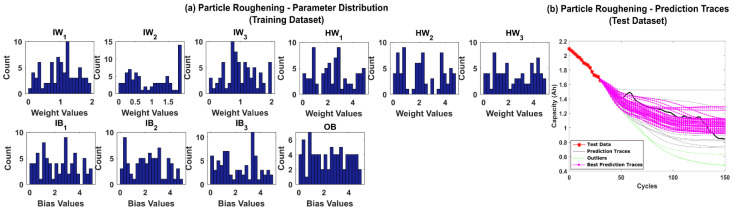
(**a**) The posterior distribution of MLP network parameters estimated by the adaptive Bayesian learning framework using particle roughening and (**b**) the degradation prediction traces for NASA battery (RW-11) using particle roughening. The prediction traces for 50 repetitions are shown using the gray lines, the green lines represent outliers, and the magenta lines represent the traces with minimum RMSE values.

**Figure 15 sensors-22-03803-f015:**
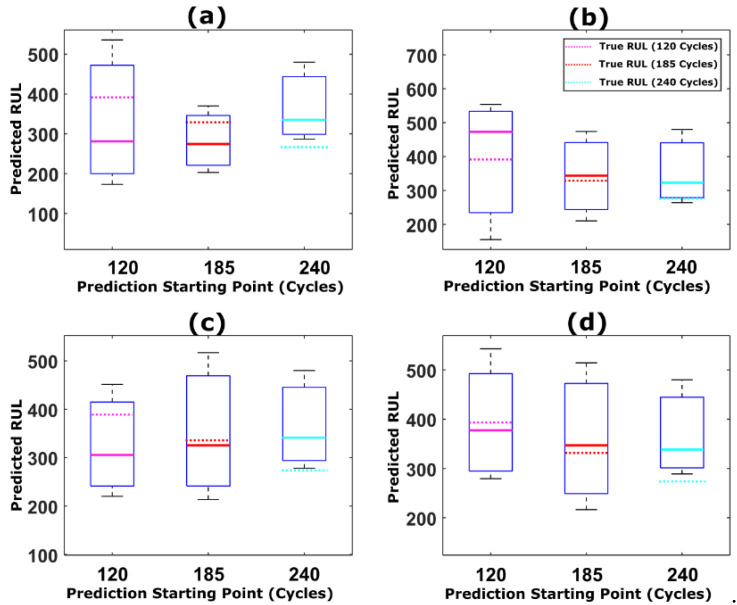
Box plot showing the comparison between predicted RUL and true RUL for CALCE dataset (CS-37) for three different prediction starting points using (**a**) MR, (**b**) StR, (**c**) SyR, and (**d**) particle roughening methods.

**Figure 16 sensors-22-03803-f016:**
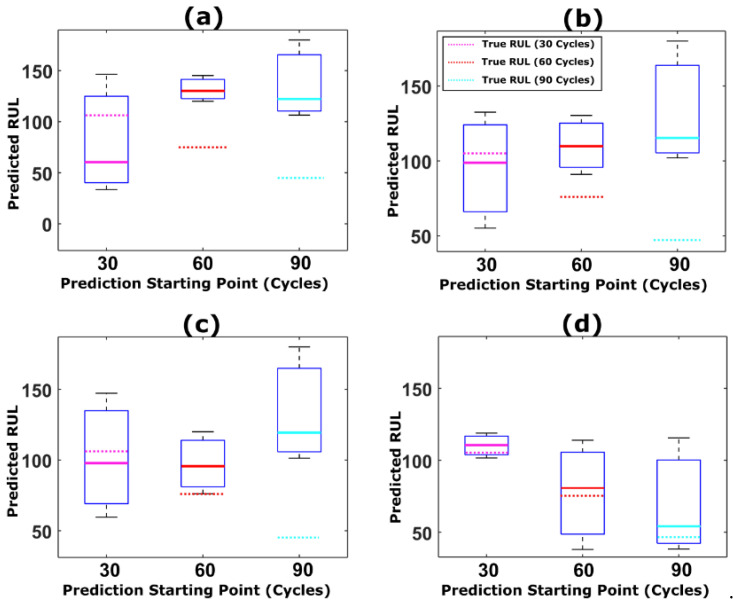
Box plot showing the comparison between predicted RUL and true RUL for NASA dataset (RW-11) for three different prediction starting points using (**a**) MR, (**b**) StR, (**c**) SyR, and (**d**) particle roughening methods.

**Table 1 sensors-22-03803-t001:** Limits of the standard deviation values chosen for particle roughening.

Sigma Label	Lower Limit (σ_r1_)	Upper Limit (σ_r2_)
Sigma—1	0.001	0.01
Sigma—2	0.01	0.1
Sigma—3	0.1	0.1

**Table 2 sensors-22-03803-t002:** Comparison of Performance Metrics for Different Resampling Strategies and Roughening for CALCE and NASA degradation datasets.

	MR				StR				SyR				Particle Roughening			
Dataset	RMSE	RA	Time (s)	Successful Iterations (%)	RMSE	RA	Time (s)	Successful Iterations (%)	RMSE	RA	Time (s)	Successful Iterations (%)	RMSE	RA	Time (s)	Successful Iterations (%)
CALCE (CS-37)	0.14	0.85	47.0	78	0.06	0.56	41.9	90	0.05	0.92	42.1	96	0.04	0.96	343	98
NASA (RW11)	0.37	0.58	66.3	84	0.35	0.72	31.3	90	0.27	0.94	83.8	98	0.24	0.95	255	94

**Table 3 sensors-22-03803-t003:** Comparison of RMSE values of the proposed method with Refs. [26,35].

Dataset	Ref. [26] (Wu et al.)	Ref. [35] (Our Previous Work)	This Work
CALCE (CS-37)	0.2698	0.050	0.040
NASA (RW11)	0.3686	0.367	0.24

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
