# Peer review of "Remaining Useful Life Prediction of Lithium-Ion Batteries Using Neural Networks with Adaptive Bayesian Learning"

_sensors, 2022, doi:10.3390/s22103803_

Round 1
Reviewer 1 Report
The manuscript entitled “Remaining Useful Life Prediction of Lithium-ion Batteries using Neural Networks with Adaptive Bayesian Learning” has been investigated in detail. The topic addressed in the manuscript is potentially interesting and the manuscript contains some practical meanings, however, there are some issues which should be addressed by the authors:
- In the first place, I would encourage the authors to extend the abstract more with the key results. As it is, the abstract is a little thin and does not quite convey the interesting results that follow in the main paper. The "Abstract" section can be made much more impressive by highlighting your contributions. The contribution of the study should be explained simply and clearly.
- The readability and presentation of the study should be further improved. The paper suffers from language problems.
- The “Introduction” section needs a major revision in terms of providing more accurate and informative literature review and the pros and cons of the available approaches and how the proposed method is different comparatively. Also, the motivation and contribution should be stated more clearly.
- The importance of the design carried out in this manuscript can be explained better than other important studies published in this field. I recommend the authors to review other recently developed works.
- “Results and Discussions” section should be edited in a more highlighting, argumentative way. The author should analysis the reason why the tested results is achieved.
- The authors should clearly emphasize the contribution of the study. Please note that the up-to-date of references will contribute to the up-to-date of your manuscript. The studies named- Crude oil time series prediction model based on LSTM network with chaotic Henry gas solubility optimization; Remaining useful life prediction of lithium-ion batteries based on false nearest neighbors and a hybrid neural network; Remaining useful life prediction of lithium-ion batteries using neural network and bat-based particle filter; Detection of solder paste defects with an optimization‐based deep learning model using image processing techniques- can be used to explain the fractal analysis and classification method in the study or to indicate the contribution in the “Introduction” section.
- The advantages of the proposed method compared with the others for the same class of problems in recent literature are not elaborated adequately. It is recommended to provide to simulation results compared with the others, which will show the performance of the proposed method in a clear manner.
- The complexity of the proposed model and the model parameter uncertainty are not enough mentioned.
- It will be helpful to the readers if some discussions about insight of the main results are added as Remarks.
This study may be proposed for publication if it is addressed in the specified problems.
Author Response
We would like to the reviewer for their valuable comments. The modifications made in the revised manuscript as per the reviewer's comments are highlighted in Yellow

Reviewer 2 Report
- Figures 11 to 15 need to be reformatted. Table 2 needs to be reformatted.
- Several error evaluation indexes need to be added to better reflect the superiority of the model proposed .
- The probability density function diagram of life prediction should be drawn in the case of remaining usefullife prediction, and different starting points should be selected for experiments. It can be seen from Figure 16 that only one prediction starting point was selected for the experiment.
- The ordinate of figure (a) in Figures 6 to 15 can change the “count” to “probability”.
- In introduction,authors should introduce up-to-date literature:Product technical life prediction based on multi-modes and fractional Lévy stable motion;Generalized Cauchy Degradation Model with Long-Range Dependence and Maximum Lyapunov Exponent for Remaining Useful Life
Author Response
Please see the attachement

Reviewer 3 Report
The authors made a huge work. Nevertheless, I have noticed several issues:
- all equations must be written in appropriate software. A Word built-in is not sufficient
- What is reliability level of each experiment? Did the authors consider it? Please, explain.
- I do not understand why authors had self-cited references 30 and 31.
I hope I help.
Round 2
Reviewer 1 Report
All my comments have been thoroughly addressed. It is acceptable in the present form.